# Guanosine-Mediated Anxiolytic-Like Effect: Interplay with Adenosine A_1_ and A_2A_ Receptors

**DOI:** 10.3390/ijms21239281

**Published:** 2020-12-05

**Authors:** Monica Frinchi, Vincenzo Verdi, Fulvio Plescia, Francisco Ciruela, Maria Grillo, Roberta Garozzo, Daniele F. Condorelli, Patrizia Di Iorio, Francesco Caciagli, Renata Ciccarelli, Natale Belluardo, Valentina Di Liberto, Giuseppa Mudò

**Affiliations:** 1Department of Biomedicine, Neuroscience and Advanced Diagnostic, University of Palermo, 90134 Palermo, Italy; Monica.Frinchi@unipa.it (M.F.); vincenzo.verdi@inserm.fr (V.V.); dott.grillomaria@gmail.com (M.G.); natale.belluardo@unipa.it (N.B.); valentina.diliberto@unipa.it (V.D.L.); 2Institut de Psychiatrie et Neurosciences de Paris, INSERM U1266, Université de Paris, F-75014 Paris, France; 3Department of Sciences for Health Promotion and Mother and Child Care “G. D’Alessandro”, University of Palermo, 90127 Palermo, Italy; fulvio.plescia@unipa.it; 4Unitat de Farmacologia, Departament Patologia i Terapèutica Experimental, Facultat de Medicina i Ciències de la Salut, IDIBELL, Universitat de Barcelona, 08907 Barcelona, Spain; 5Institut de Neurociències, Universitat de Barcelona, 08035 Barcelona, Spain; 6Department of Biomedical and Biotechnological Sciences, Section of Medical Biochemistry, University of Catania, 95123 Catania, Italy; robertagaroz@hotmail.it (R.G.); daniele.condorelli@unict.it (D.F.C.); 7Laboratory of Complex Systems, Scuola Superiore di Catania, University of Catania, 95123 Catania, Italy; 8Department of Medical, Oral and Biotechnological Sciences, University of Chieti-Pescara, 66100 Chieti, Italy; patrizia.diiorio@unich.it (P.D.I.); f.caciagli@unich.it (F.C.); renata.ciccarelli@unich.it (R.C.); 9Center for Advanced Studies and Technologies (CAST), University of Chieti-Pescara, 66100 Chieti, Italy

**Keywords:** guanosine, adenosine, behavior, A_1_R, A_2A_R, caffeine

## Abstract

Acute or chronic administration of guanosine (GUO) induces anxiolytic-like effects, for which the adenosine (ADO) system involvement has been postulated yet without a direct experimental evidence. Thus, we aimed to investigate whether adenosine receptors (ARs) are involved in the GUO-mediated anxiolytic-like effect, evaluated by three anxiety-related paradigms in rats. First, we confirmed that acute treatment with GUO exerts an anxiolytic-like effect. Subsequently, we investigated the effects of pretreatment with ADO or A_1_R (CPA, CCPA) or A_2A_R (CGS21680) agonists 10 min prior to GUO on a GUO-induced anxiolytic-like effect. All the combined treatments blocked the GUO anxiolytic-like effect, whereas when administered alone, each compound was ineffective as compared to the control group. Interestingly, the pretreatment with nonselective antagonist caffeine or selective A_1_R (DPCPX) or A_2A_R (ZM241385) antagonists did not modify the GUO-induced anxiolytic-like effect. Finally, binding assay performed in hippocampal membranes showed that [^3^H]GUO binding became saturable at 100–300 nM, suggesting the existence of a putative GUO binding site. In competition experiments, ADO showed a potency order similar to GUO in displacing [^3^H]GUO binding, whereas AR selective agonists, CPA and CGS21680, partially displaced [^3^H]GUO binding, but the sum of the two effects was able to displace [^3^H]GUO binding to the same extent of ADO alone. Overall, our results strengthen previous data supporting GUO-mediated anxiolytic-like effects, add new evidence that these effects are blocked by A_1_R and A_2A_R agonists and pave, although they do not elucidate the mechanism of GUO and ADO receptor interaction, for a better characterization of GUO binding sites in ARs.

## 1. Introduction

In the last two decades, guanine-based purines (GBPs), particularly guanosine (GUO), have been shown to exert extracellular effects through putative membrane receptor(s) affecting several cellular processes, including neuronal growth, differentiation and survival [1,2,3,4,5]. These GUO effects may be mediated through putative G protein-coupled receptors (GPCRs) whose signaling might involve cyclic nucleotides or MAP kinase pathways [6,7]. However, though GUO binding sites activating putative GPCRs at rat brain membranes have been reported [8,9], at present, GUO remains an orphan ligand.

In addition to the above-mentioned data showing the existence of a putative unknown GPCR for GBPs, other findings indicated that GUO may signal through the adenosine (ADO) receptor (AR) [1,3,10]. The ARs family includes four receptors subtypes: A_1_R, A_2A_R, A_2B_R and A_3_R [11]. A_1_R and A_2A_R are the major ARs subtypes expressed in the brain, with A_1_Rs widely distributed in the cortex, hippocampus and cerebellum, whereas A_2A_Rs mostly expressed in the striatum and olfactory bulb [11]. In contrast, A_2B_R and A_3_R are weakly expressed in the brain. The A_1_R and A_2A_R possess high affinity for ADO, while the A_2B_R and A_3_R show relatively lower affinity. In neurons, both A_1_R and A_2A_R are highly localized to synaptic regions, where they modulate the release of neurotransmitters, such as glutamate, acetylcholine, serotonin and gamma-aminobutyric acid (GABA) [12]. In the brain ADO integrates inhibitory and excitatory neurotransmission mainly acting on A_1_R and A_2A_R subtypes, respectively [13].

Recently, several studies have reported the involvement of ARs in GUO-mediated behavioral outcomes, such as anxiolytic-like effects or inhibition of anxiety manifestations [14,15]. Anxiety disorders, such as panic disorder, post-traumatic stress disorder and generalized anxiety disorder, are among the most prevalent and disabling psychiatric disorders worldwide. Anxiety is characterized by several symptoms, such as tachycardia, sweating, dyspnea, fear, discomfort or psychological stress [16]. The pathophysiological mechanisms underlying anxiety involve different neurotransmitters, mainly including GABAergic system [17]. Indeed, benzodiazepines, which act by increasing the inhibitory GABAergic neurotransmission, are currently used as treatment for anxiety. However, other neurotransmitters such as ADO, can mediate anxiogenic and anxiolytic actions. Indeed, mice lacking A_1_R [18,19] or A_2A_R [20] display enhanced anxiety, while A_1_R agonist N6-Cyclopentyladenosine (CPA) is able to mediate an anxiolytic-like effect dependent on the dose used [21,22]. Interestingly, acute or chronic administration of GBPs produces an anxiolytic-like effect [3,14,15,23,24].

Indeed, Guanosine-5′-monophosphate (GMP) has been shown to be able to induce anxiolytic-like behavior mediated by a reduction in glutamatergic transmission [25]. Additionally, acute or chronic administration of GUO produced an anxiolytic-like effect as evaluated by several paradigms, such as the hole board task [15], the tail suspension test, the open field test (OFT) [3,23] or the elevated plus maze (EPM) test [14] and more recently through different behavioral tests [24]. These anxiolytic-like effects produced by GUO treatment were mainly correlated with glutamate release inhibition, which in turn was mediated by enhancement in extracellular release of ADO [14] and involvement of A_1_R or A_2A_R [14,24,25]. Indeed, recent data reported that GUO-mediated anxiolytic-like effects were prevented by A_1_R antagonist (DPCPX; 8-Cyclopentyl-1,3-dipropylxanthine) [14,24] or by the ARs antagonist caffeine [14], even though other data showed that caffeine did not block them [26]. Accordingly, the research reporting the involvement of ARs in GUO-mediated behavioral outcomes is still ambiguous and remains to be elucidated.

Since several findings supported the notion that both A_1_R and A_2A_R are primarily responsible for the central effects of ADO [27], and would mainly participate in GUO-mediated effects [1,3,10], including anxiolytic-like effects [14,24,25], in the present work, we aimed validate the involvement of A_1_R and A_2A_R in the GUO-mediated anxiolytic-like effect. To this end, by using three anxiety-related paradigms in rats, we first confirmed the GUO anxiolytic-like effect after acute in vivo treatment, according to the data reported by Almeida et al. [14]. Then, we assessed whether the administration of A_1_R and A_2A_R agonists or antagonists combined with GUO would result in functional competition. Finally, since years ago a binding site for GUO was detected on membrane preparations from rat brain [8,9], we also verified the existence of binding site competition between GUO and AR agonists or antagonist in hippocampal membranes.

## 2. Results

### 2.1. Elevated Plus Maze Test

We first used EPM test to gauge the dose and time-point affording the maximum GUO-mediated anxiolytic effect, according to the treatment scheme inserted in Appendix A. Thus, the evaluation of dose-effect experiments showed that 10 mg/kg GUO had no anxiolytic effects, whereas the dose of 20 mg/kg significantly increased the entries in open arms (Figure 1A), but not the time spent in open arms and the ratio between time spent in open arms and total time in arms (Figure 1B,C). Interestingly, treatment with 30 mg/kg GUO significantly increased all the above mentioned parameters (Figure 1A–C), thus showing the best anxiolytic-like effect. The evaluation of the time-course of the GUO-mediated anxiolytic-like effect showed that treatment with 30 mg/kg of GUO significantly increased at 45 min the entries (Figure 1D), whereas it significantly increased even at 180 min the time in open arms (Figure 1E), including the ratio of time in open arms and total time in arms (Figure 1F).

Next, we assessed the impact of nonselective ARs agonist ADO or A_1_R and A_2A_R selective agonists (i.e., CPA, CCPA and CGS21680, respectively) treatment in the GUO-mediated anxiolytic-like effects (Appendix A). Interestingly, pretreatment with 10 mg/kg of ADO resulted in a substantial inhibition of the anxiolytic-like effect of GUO with a significant decrease in the time spent in open arms and the ratio between the time spent in open arms and the total time in arms (Figure 2A–C). Interestingly, pretreatment with 30 mg/kg of ADO produced not only a complete block of GUO anxiolytic-like effect in all EPM test parameters but also caused a significant reduction in the entries in open arms and the ratio between the time spent in open arms and the total time in arms when compared to the control group (Figure 2A–C), suggesting the induction of an anxiogenic effect. However, administration of ADO alone showed no significant change in any EPM test parameters as compared to the control group (Figure 2A–C), thus indicating that the doses of ADO used were neither anxiogenic nor anxiolytic.

Pretreatment with A_1_R agonist CPA (0.1 mg/kg) completely blocked the anxiolytic-like effect of GUO in all EPM test parameters monitored (Figure 2D–F). Comparable results were obtained by pretreatment with another A_1_R agonist CCPA (0.1 mg/kg) (Appendix A). Subsequently, pretreatment with A_2A_R agonist CGS21680 (0.4 mg/kg) caused a complete block of the GUO-effect in all EPM test parameters (Figure 2D–F), with even significant reduction in time spent in open arms when compared to the control group (Figure 2E), thus suggesting an anxiogenic effect. Interestingly, treatment with CGS21680 alone significantly reduced the time spent in open arms and the ratio of time in open arms and total time in arms as compared to the control group (Figure 2E,F), thus suggesting again an anxiogenic effect of the A_2_AR agonist. All agonists at the dose used did not induce obvious motor disturbance—e.g., sedative effects—as verified in the OFT test (data not shown).

Next, we aimed to verify whether a canonical nonselective ARs antagonist caffeine had any impact on the GUO-mediated anxiolytic-like effect (Appendix A). Interestingly, caffeine alone induced a significant anxiolytic-like effect, involving both the time spent in open arms and the ratio of time in open arms and total time in arms (Figure 3B,C). When caffeine was administered before GUO, the anxiolytic-like effect observed, involving the time spent in open arms and the ratio of time in open arms and total time in arms, was like that of the individual drugs (Figure 3A–C). This indicates that combined caffeine and GUO treatment did not show an additive anxiolytic-like effect. In analogy to the functional competition between caffeine and GUO, we next verified the functional competition between caffeine and ADO, performing pretreatment with caffeine. However, the presence of ADO did not modify the anxiolytic-like effect of caffeine, as shown by the preservation of caffeine significant effect in time spent in open arms and ratio of time in open arms and total time in arms in comparison to the control group (Figure 3B,C).

Subsequently, we examined the effects of A_1_R and A_2A_R selective antagonists on the anxiolytic-like effect of GUO (Appendix A). Importantly, pretreatment with DPCPX, a selective A_1_R antagonist, or ZM241385, a selective A_2A_R antagonist, was unable to counteract the GUO anxiolytic-like effect, involving the time spent in open arms and the ratio of time in open arms and total time in arms (Figure 3D–F). Finally, DPCPX or ZM241385 treatment alone did not produce anxiolytic-like effect as compared to the control group (Figure 3D–F).

### 2.2. Light–Dark Box and Open Field Tests

To further validate the GUO-mediated anxiolytic-like effects, we next implemented two EPM complementary anxiety-related paradigms, namely light–dark box (LDB) and OFT (Appendix A). Indeed, in the LDB test, GUO (30 mg/kg) treated animals showed a significant increase in time spent in the light box as compared to the control group (Figure 4A,B), thus suggesting again a GUO-mediated anxiolytic-like effect. Interestingly, this anxiolytic effect of GUO was completely blocked by pretreatment with nonselective ARs agonist ADO, whereas ADO alone did not affect behavior as compared to the control group (Figure 4A,B). Conversely, in the OFT test, GUO treatment (30 mg/kg) significantly increased both the number of central transitions and the amount of time spent in the center of the arena as compared to the control group (Figure 4C,D). Again, these anxiolytic effects of GUO were completely blocked by pretreatment with ADO. When administered alone, ADO did not affect the rat behavior as compared to the control group (Figure 4C,D). Importantly, a close analysis of the OFT test results revealed that the GUO treated animals, when compared to the control group, increased the total distance travelled (36%), the path length in center of the arena (67%; *p* < 0.01) and the total mobility time (35%) (Appendix A). In addition, ADO pretreatment precluded GUO effects and decreased the time of total mobility (−86%; *p* < 0.01), the total distance travelled (−55%; *p* < 0.05) and the path length in center (−63%; *p* < 0.05) (Appendix A). ADO treatment alone did not produce any significant change in all these parameters (Appendix A). Overall, these results indicated that no obvious motor disturbance such as sedative effects were induced after acute administration of GUO or ADO.

### 2.3. [^3^H]Guanosine Binding to Hippocampal Membranes

In order to correlate the behavioral effects of GUO to a putative receptor site in rat brain, we investigated the binding of [^3^H]GUO to hippocampal membranes (Figure 5). Indeed, we selected the hippocampus for our radioligand binding experiments in view of its role in controlling mood and anxiety. The saturation isotherm studies showed that the binding became saturable at [^3^H]GUO concentrations ranging between 100 and 300 nM (Figure 5A). The pooled data were fitted by a computerized nonlinear regression analysis and resolved for the presence of a single high binding site with an apparent K_D_ of 80 ± 34 nM and a B_max_ of 2339 ± 339 fmol/mg protein (Figure 5A). Overall, these results demonstrated the existence of a putative GUO binding site in hippocampal membranes.

Subsequently, we aimed to compare GUO and ADO binding kinetics through competition binding assays. To this end, the ability of increasing concentrations (i.e., 1nM to 1 mM) of GUO and ADO to displace 70 nM [^3^H]GUO binding was assessed. Interestingly, the data showed that ADO and GUO exhibited almost the same potency order [IC_50_ 85nM (log IC_50_ −6.069 ± 0.2074) and 56nM (log IC_50_ −6.251 ± 0.1649), respectively] to displace [^3^H]GUO binding (Figure 5B).

Finally, we measured the ability of 500 µM of GUO, ADO, CPA, CGS21680 or caffeine to displace specifically 70 nM [^3^H]GUO binding in rat hippocampal membranes. As shown in Figure 5C, ADO with 68.6% of displacement was almost as effective as GUO (87.5%) in displacing [^3^H]GUO binding. A_1_R agonist CPA was able to displace 56.9% of [^3^H]GUO binding, whereas A_2A_R agonist CGS21680 displaced only 11% of [^3^H]GUO binding (Figure 5C). Interestingly, taken together CPA and CGS21680 were able to displace [^3^H]GUO binding to the same extent of ADO alone. Caffeine displaced only 24% of [^3^H]GUO binding (Figure 5C).

## 3. Discussion

While the effects of GUO on animal behavior has been largely demonstrated [15,23], the existence of GUO receptors in the brain is still an open question. Here, we demonstrated, using a model of anxiolytic-like effect induced by acute GUO treatment in naïve rats [14], the existence of functional competition between A_1_R and A_2A_R agonists or antagonists and GUO. An anxiolytic-like effect has been previously reported upon GMP or GUO administration, an effect relying on the modulation of the glutamatergic system [14,15,23,25]. In addition, antidepressant-like effects have been reported following acute or chronic administration of GUO [28,29], again correlating well with the modulation of glutamatergic neurotransmission [1,2,3]. Here, we demonstrated that the GUO-mediated anxiolytic-like effect was blocked by pretreatment with ARs nonselective agonist ADO or A_1_R -A_2A_R related selective agonists at doses that did not produce anxiolytic responses per se, thus suggesting a cross-talk between GUO and ADO through putative adenosine-related receptors. Indeed, in our hands, the anxiolytic-like effect of GUO in the EPM test was blocked by A_1_R and A_2A_R agonists but not by A_1_R and A_2A_R antagonists, thus correlating with the fact that A_2A_R activation, but not its blockade, inhibits GUO-induced glutamate uptake [30]. In line with the hypothesis of the involvement of ADO receptors in GUO-mediated anxiolytic-like effects, it has been recently reported that GUO attenuation of behavioral deficits after traumatic brain injury can be modulated by ADO receptors, particularly A_1_R [24]. Recently, the notion that ARs appear to be involved in the pathophysiology of mood disorders has been suggested [31]. Particularly, selective activation of A_1_R induces anxiolytic-like behavior, while the activation of A_2A_R sites reduces the effects of A_1_R activation [21]. In addition, positive allosteric modulation of A_1_R produces anxiolytic-like effects [32] or mice lacking A_1_R [18,19,33] or A_2A_R [20] display enhanced anxiety. A_1_R agonist CPA mediates anxiolytic-like effect dependent on the dose used [21,22], and here, we used CPA at doses that did not produce anxiolytic responses per se.

Interestingly, here, we evidenced that [^3^H]GUO binding to hippocampal membranes may be efficiently displaced by ADO and by the A_1_R selective agonist CPA. Indeed, ADO and selective A_1_R agonist CPA displaced 70% and 57% of [^3^H]GUO binding, respectively. This result was in agreement with present functional data showing a competition binding between ADO or A_1_R agonist CPA and GUO. In contrast, the selective A_2A_R agonist CGS21680 displaced only 11% of [^3^H]GUO binding. Surprisingly, while the A_2A_R selective agonist CGS21680 showed a potent inhibition of GUO-mediated anxiolytic-like effect, it produced a very low [^3^H]GUO displacement. However, this different percentage of [^3^H]GUO displacement between A_1_R and A_2A_R agonists may find explanation in the more prevalent amount of A_1_R than A_2A_R in the hippocampus [34]. In support of this possibility, taken together, CPA and CGS21680 were able to displace [^3^H]GUO binding to the same extent as ADO alone. The same explanation might also support the apparent involvement of A_1_R and not A_2A_R in GUO attenuation of behavioral deficits after traumatic brain injury [24]. Indeed, since A_1_R and A_2A_R are engaged in heteroreceptor complexes in the cell membrane [35], both the receptors are likely involved in GUO effects, as recently reported by Lanznaster et al. [10].

In the same experimental conditions, we found a very low level of [^3^H]GUO binding displacement with caffeine (24%), and this result was in good correlation with functional data showing that caffeine and selective A_1_Rs and A_2A_Rs antagonists were not able to block the anxiolytic-like effect of GUO. This low competition between caffeine and [^3^H]GUO binding could further suggest that GUO might interact with ARs sharing binding sites with ADO.

In fact, specific receptor binding sites for guanosine triphosphate (GTP) in PC12 cells [36] and for GUO in rat brain membranes have been described [8,37], supporting the possible existence of putative membrane receptors [38]. This GUO binding on rat brain membranes was not displaced by ADO and caffeine [37] in contrast with the present data showing that ADO has almost the same potency order of the GUO to displace [^3^H]GUO. However, our data are strengthened by the functional outcome and are confirmed by [^3^H]GUO displacement with selective AR agonists.

Although not without controversy, an interplay between GBP effects and ARs activity has been postulated for a long time and recently reviewed. In brief, several results indicated that GUO may signal through A_1_R and/or A_2A_R involvement [30,39,40,41,42], whereas other evidence shows that GUO effects seem to be independent from A_1_R and A_2_R activation [15,26,43].

Here, we observed a functional competition between GUO and ADO or A_1_R-A_2A_R selective agonists, which might be supported by two potential hypotheses:

(1) Competition binding site: in ARs, there is a binding site for GUO that might generate an allosteric modulation or alternatively putative GUO receptors might share some features with ARs. In the present study, the anxiolytic-like effect of GUO blocked by the pretreatment of rats with ADO and with A_1_R or A_2A_R selective agonists may suggest a competition binding between ADO and GUO in ARs or alternatively an allosteric modulation between ADO and GUO binding sites in the ARs. A third possibility is the existence of putative GUO receptors sharing features with ARs that could directly bind ADO or AR agonists, as supported by present data showing [^3^H]GUO binding displaced by ADO and by A_1_R agonist. The AR nonselective antagonist caffeine did not antagonize the anxiolytic-like effect of GUO, and this result was in good correlation with the lack of [^3^H]GUO binding displacement by caffeine in hippocampal membranes. However, this result was in contrast with previous data showing that caffeine may block the anxiolytic-like behavior induced by GUO in rats [14]. Moreover, in our experiments, treatment with caffeine alone showed an anxiolytic-like effect [15,44,45,46], and co-treatment with GUO and caffeine showed, surprisingly, less anxiolytic-like effect than that induced by GUO or by caffeine alone.

All together, the present data may suggest that the GUO-mediated anxiolytic-like effect may be blocked by AR activation and that GUO, probably with low affinity [47], may bind to ARs in allosteric manner and may act as negative modulator or as agonist of ARs, triggering alternative pathways to those promoted by ADO. As is well known, allosteric modulators may exert different effects on agonist vs. antagonist binding mode [48].

(2) Receptor–receptor interaction: GUO putative receptors may form heteroreceptor complexes with ARs modulating the reciprocal activity. The functional competition observed between GUO and ADO in the modulation of the anxiolytic-like effect of GUO also suggested the hypothesis that there is an unknown specific receptor for GUO that may form heterocomplexes with ADO receptors generating a reciprocal modulation [1,49,50,51]. The existence of putative GBP receptors in the brain is still a deduction of several experimental cellular effects [52,53]. By monitoring the binding of a nonhydrolysable-labeled GTP in rat brain membranes, it has also been demonstrated that GUO is able to activate a putative not yet identified GPCR [9]. We also showed the activation of a putative unknown GPCR for GUO by means of binding experiments and in situ autoradiography of [^35^S]GTPγS, respectively in membranes and slices from rat brain [54]. Based on the evidence that ADO receptors, mainly A_1_R and A_2A_R, form heteroreceptor complexes with several P2Y receptor families [55], it is reasonable to hypothesize a possible interaction between a putative GUO receptor and ARs, according to a previously reported hypothesis [3,56]. Recently, in partial support of this hypothesis, an important role for the A_1_R-A_2A_R receptor–receptor interaction in GUO-mediated effects was reported by Lanznaster et al. [3].

## 4. Materials and Methods

### 4.1. Animals and Drugs

Animals. Rats were housed in a specific pathogen-free environment, three per polypropylene cages in controlled temperature (23 ± 2 °C), humidity (50–55%) and light (12-h light/dark cycle), with access to food and water ad libitum. The experiments were carried out in accordance with the National Institute of Health Guidelines for the Care and Use of Mammals in Neuroscience and Behavioral Research (The National Academics Press, WA, USA), with the rules and principles of the European Communities Council Directive 2010/63/EU revising Directive 86/609/EEC, in accordance with the national D.L. March 4, 2014, no. 26, and were approved by the local Animal Care Committee (OPBA) of University of Palermo, Italy and Ministry of Health, Italy. No other methods to perform the described experiments (3Rs) were found.

Drugs. The following drugs were used: GUO (G6264 Sigma-Aldrich St. Louis, MO, USA) and nonselective ARs ligand ADO (A3377 Sigma-Aldrich St. Louis, MO, USA) were dissolved in 0.9% physiological saline solution (pH 9.0); selective A_1_R agonists N6-Cyclopentyladenosine (CPA, C8031 Sigma-Aldrich) and 2-Chloro-N6-cyclopentyladenosine (CCPA, 1705 Tocris) or A_2_AR agonist 2-p-(2-Carboxyethyl)phenethylamino-5′-*N*-ethylcarboxamidoadenosine hydrochloride (CGS21680 hydrocloride, sc-211062 Santa Cruz Biotechnology) as well as selective A_1_R antagonist 8-Cyclopentyl-1,3-dipropylxanthine (DPCPX, sc-200115 Santa-Cruz Biotechnology) or A_2A_R antagonist 4-(-2-[7-amino-2-{2-furyl}triazolo [57] triazin-5-yl-amino]ethyl)phenol (ZM241385, Z0153, Sigma-Aldrich) were first dissolved in DMSO and thereafter in 0.9% physiological saline solution; the ARs nonselective antagonist caffeine (C0750, Sigma-Aldrich) was dissolved in 0.9% physiological saline solution. All drugs were administered intraperitoneally (i.p.) in a volume of 0.2 mL.

### 4.2. Behavioral Tests

Three different anxiety-related paradigms were used in the present study: an elevated plus maze (EPM) test, open field test (OFT) and light/dark box (LDB) test [58]. Each behavior testing, lasting 5 min, was performed between 8:30 and 13:30 a.m. and at a scheduled time of 45 min after drug treatment (Appendix A).

Elevated plus maze test. The EPM test was conducted as described by Di Liberto et al. [58]. Briefly, each animal was placed on the central platform facing one of the open arms, in a mean light intensity (100 lx) illuminated chamber. During a 5-min test period, the following parameters were recorded: (a) number of open arm entries; (b) number of closed arm entries; (c) time spent in open arms; (d) time spent in closed arms; (e) ratio time in open arms and total time in arms. The behavior in the maze was recorded using a computerized video tracking system (Any-Maze software v. 4).

Open field test. The OFT was performed in an open field arena using the Opto-Varimex system (Columbus Instruments, Columbus, OH, USA). This apparatus was a square box—44 cm wide, 44 cm long and 20 cm high—whose perpendicular sides had 15 infrared emitters. Testing was performed in a mean light intensity (100 lx) illuminated chamber, and each animal was placed in the same corner of the open field arena and its behavior was recorded for 5 min. The variables observed were: (a) the first latency to enter the central zone of the open field arena; (b) the number of entries in the central zone of the open field arena; (c) the amount of the time spent in the central zone.

Light–dark box test. The light–dark box (LDB) apparatus consisted of a wooden box (60 cm length × 30 cm height × 30 cm width) divided into two equal-size compartments by a barrier that contained a doorway (10 cm height × 10 cm width). One of the compartments was painted black and was covered with a lid and the other compartment was painted white and illuminated with a 45 W light bulb positioned 40 cm above the box. The rats were placed in the middle of the lit compartment and allowed to freely explore the apparatus for 5 min. The number of transitions and time spent in the light compartment were measured.

### 4.3. Behavioral Experimental Design

Adult male Wistar rats (4 months old) were used. All experimental groups consisted of at least eight rats. To define the dose-effect and the time-course of GUO-mediated anxiolytic-like effects, we first performed an EPM test. Thus, for the dose-effect study, four experimental groups were used: (i) control (vehicle); (ii) GUO (10 mg/Kg); (iii) GUO (20 mg/Kg); (iv) GUO (30 mg/Kg). Four experimental groups were used for the time-course study, namely 20, 45, 90 and 180 min, using GUO (30 mg/Kg). Once the optimal dose and time-point of the GUO-mediated anxiolytic-like effect were defined, we performed competition experiments between GUO and nonselective ARs agonist ADO using six groups of rats: (i) control (vehicle); (ii) GUO (30 mg/Kg); (iii) ADO (10 or 30 mg/kg) given 10 min before GUO (30 mg/kg); iv) ADO (10 or 30 mg/kg). Six groups of rats were used for competition experiments between GUO and selective A_1_R or A_2A_R agonists (i.e., CPA and CGS21680, respectively): (i) control (vehicle); (ii) GUO (30 mg/kg); (iii) CGS21680 (0.4 mg/kg) or CPA (0.1 mg/kg) given 10 min before GUO (30 mg/kg); (iv) (0.4 mg/kg) or CPA (0.1 mg/kg). CPA and CGS21680 doses were chosen based on preliminary investigations showing anxiolytic-like effects associated with CPA 0.2 mg/Kg, and according to literature data that excluded anxiolytic response associated to acute treatment with CPA 0.1 mg/Kg and CGS21680 0.4 mg/Kg [21,59].

In order to deeply explore the competition between GUO and another selective A_1_R agonist (CCPA), an extra experiment was performed, by using four groups of rats: (i) control (vehicle); (ii) GUO (30 mg/kg); (iii) CCPA (0.1 mg/kg) given 10 min before GUO (30 mg/kg); (iv) CCPA (0.1 mg/kg). CCPA 0.1 mg/Kg dose was chosen based on literature data, reporting no anxiolytic effects associated to this dose [57,60].

Eight groups of rats were used for competition experiments using nonselective ARs antagonists (i.e., caffeine) or selective A_1_R and A_2A_R antagonists (i.e., DPCPX and ZM241385, respectively): (i) control (vehicle); (ii) GUO (30 mg/kg); (iii) caffeine (30 mg/kg), ZM241385 (0.1 mg/kg) or DPCPX (1mg/kg) 10 min before GUO (30 mg/kg); (iv) caffeine (30 mg/kg), ZM241385 (0.1 mg/kg) or DPCPX (1mg/kg). Finally, the impact of ADO in GUO-mediated anxiolytic-like response was further investigated with OFT and LDB behavioral tests using the following groups of rats: (i) control (vehicle); (ii) GUO (30 mg/Kg); (iii) ADO (30 mg/kg) given 10 min before GUO (30 mg/kg); (iv) ADO (30 mg/kg) (see Appendix A).

### 4.4. Hippocampal Membranes Preparation

The hippocampal brain region was dissected and homogenized in ice-cold Tris-EGTA buffer (50 mM Tris-HCl, 1 mM EGTA, 3 mM MgCl_2_, 100 mM NaCl, pH 7.4) using an Ultra-turrax TP 18/10 instrument (Janke and Kunkel, IKA Werk Staufen im Breisgau, Germany) for 5 s at maximum setting level. The homogenate was sonicated (30 pulsations/min) and centrifuged at 600 g, for 5 min at 4 °C. The supernatant was centrifuged at 16170 xg for 20 min at 4 °C in a Centra MP4R refrigerated centrifuge with an 851(651) rotor. The obtained pellet, containing membrane fraction, was re-suspended in ice-cold buffer plus protease inhibitor cocktail (P8340, Sigma Aldrich) and quantified by the Lowry method [61].

### 4.5. [^3^H]GUO Binding Assay in Hippocampal Membranes and Competition Studies

Hippocampal membranes (50 μg) were preincubated for 10 min at 30 °C in binding buffer (20 mM Tris-HCl, 1 mM EGTA, 5 mM MgCl_2_, 100 mM NaCl, pH 7.4) and then incubated for 30 min at 30 °C with [^3^H]guanosine ([^3^H]GUO) in a total volume of 0.5 mL binding buffer. For saturation experiments, a concentration range of 5–500 nM ([^3^H]GUO) was used. Nonspecific binding was determined by the addition of 500 μM unlabeled GUO. Specific binding was calculated by subtracting nonspecific from total binding. In competition experiments, displacing agents at different concentrations and 70 nM [^3^H]GUO were added, and the reaction was started by adding the membranes. After 30 min, the reaction was stopped by adding 3 mL of cold binding buffer, and the samples were rapidly filtered by vacuum filtration using Whatman GF/B glass fiber filters. Filters were washed four times with 2.5 mL cold binding buffer each time, dried for 1 h at 30 °C, transferred in scintillation vials and immersed in 5 mL of Filter count scintillation cocktail (Beckman). Bound radioactivity was measured in a Beckman Coulter LS6500 Multipurpose Scintillation Counter. For saturation and displacement curves, the pooled data were fitted by a computerized nonlinear regression analysis and resolved for the presence of a single high affinity binding site.

### 4.6. Statistical Analysis

Data are the results of the average of 6–8 animals for each experimental group. Statistical analyses were performed using one-way analysis of variance (ANOVA). Significant effects were further evaluated by the Tukey post hoc test. Values are shown as the mean ± SD. Differences in *p*-value less than 0.05 were considered statistically significant. The statistical comparison showed in the figures are restricted only to those relevant for the purpose of the study. All the behavioral data in the figures are expressed as a control percentage.

## 5. Conclusions

Overall, our results strengthen previous data supporting the anxiolytic-like effect of GUO and add new evidence that GUO may exert its anxiolytic-like effect through interaction with both A_1_R and A_2A_R.

This anxiolytic-like effect of GUO can be blocked by A_1_R and A_2A_R agonists but not by A_1_R and A_2A_R antagonists, as also supported by data of [^3^H]GUO competition binding with ADO and A_1_R and A_2A_R selective agonists to hippocampal membranes. Although these findings do not elucidate the mechanism involved, they pave the way for a better characterization of GUO binding sites in ARs.

## Figures and Tables

**Figure 1 ijms-21-09281-f001:**
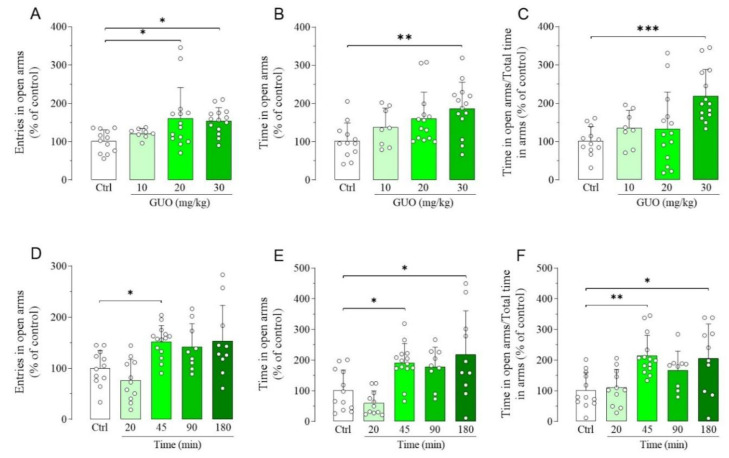
Dose-effect and time-course of guanosine (GUO)-mediated anxiolytic-like effect evaluated by elevated plus maze (EPM) test during a 5 min session. Dose-effect (GUO 45 min): (**A**) number of open arm entries [F_(3,44)_ = 3.929, *p* < 0.02]; (**B**) time spent in open arms [F_(3,44)_ = 4.441, *p* < 0.01]; (**C**) ratio time in open arms and total time in arms [F_(3,44)_ = 6.939, *p* < 0.001]. (**D**–**F**) Time-course (30 mg/kg GUO): (**D**) number of open arm entries [F_(4,51) =_ 6.710, *p* < 0.0005]; (**E**) time spent in open arms [F_(4,51)_ = 7.672, *p* < 0.0001]; (**F**) ratio time in open arms and total time in arms [F_(4,51)_ = 6.148, *p* < 0.0005]. Ctrl: Control group. Each bar represents the mean value ± SD. Tukey test: * *p* < 0.05, ** *p* < 0.01, *** *p* < 0.001. Ctrl raw mean values: (**A**) 6.75; (**B**) 60.05 s; (**C**) 0.27; (**D**) 8.58; (**E**) 95.84 s; (**F**) 0.45.

**Figure 2 ijms-21-09281-f002:**
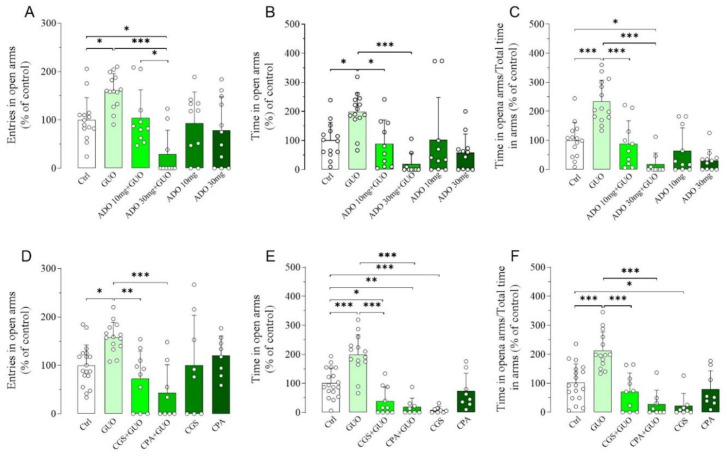
Functional competition between GUO and adenosine (ADO) or selective agonists for A_1_R and A_2A_R on GUO-mediated anxiolytic-like effect evaluated by EPM test during a 5 min session. Pretreatment with ADO (10 mg/kg or 30 mg/kg) resulted in a complete block of anxiolytic-like effect of GUO (30 mg/kg) with significant decrease in (**A**) number of open arm entries [F_(5,62)_ =6.972, *p* < 0.0001], (**B**) time spent in open arms [F_(5,62)_ = 6.566, *p* < 0.0001] and (**C**) the ratio of time in open arm and total time in arms [F_(5,62)_ = 18.33, *p* < 0.0001], when compared to GUO treated group. Pretreatment with A_1_R agonist CPA (0.1 mg/kg) or with A_2A_R agonist CGS (0.4 mg/kg) was able to completely block the anxiolytic-like effect of GUO (30 mg/kg) with significant decrease in (**D**) number of open arm entries [F_(5,60)_ = 5.450, *p* < 0.0005], (**E**) time spent in open arms [F_(5,60)_ = 22.29, *p* < 0.0001] and (**F**) the ratio of time in open arm and total time in arms [F_(5,60)_ = 15.54, *p* < 0.0001], when compared to GUO treated group. CGS alone produced a significant reduction in time spent in open arms and ratio of time in open arm and total time in arms as compared to control group (Ctrl): each bar represents the mean value ± SD. Tukey test: * *p* < 0.05, ** *p* < 0.01; *** *p* < 0.001. Ctrl raw mean values: (**A**) 6.23; (**B**) 76.23 s; (**C**) 0.33; (**D**) 7; (**E**) 97.88 s; (**F**) 0.43.

**Figure 3 ijms-21-09281-f003:**
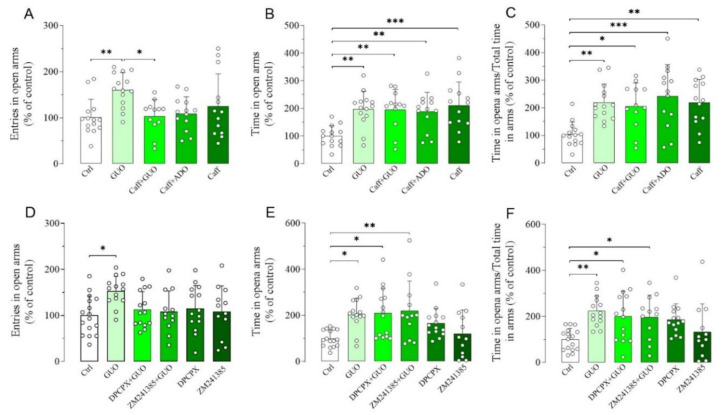
Effects of A_1_R and A_2A_R antagonists on anxiolytic-like effect of GUO evaluated by EPM test during a 5 min session. (**A**–**C**) Functional competition between GUO and nonselective ARs antagonist caffeine (Caff). Caffeine alone (30 mg/kg) induced anxiolytic effects, as shown by significant increase in time spent in open arms and ratio of time in open arms and total time in arms as compared to control group (Ctrl). Pretreatment with caffeine (30 mg/kg) did not block anxiolytic-like effect of GUO and showed significant increase in time in open arms and ratio of time in open arms and total time in arms as compared to control group. (**A**–**C**) Functional competition between caffeine (30 mg/kg) and ADO (30 mg/kg) showed preservation of caffeine anxiolytic effect with significant increase in time spent in open arms and ratio of time in open arms and total time in arms as compared to control group. (**D**–**F**) Functional competition between GUO and selective AR antagonists. Selective A_1_R antagonist DPCPX (1 mg/kg) was not able to block GUO anxiolytic-like effect, as shown by significant increase in time in open arms and ratio of time in open arms and total time in arms as compared to control group. DPCPX alone did not show significant changes of all EPM parameters as compared to control group. Pretreatment with selective A_2A_R antagonist ZM241385 (0.4 mg/kg) was not able to block GUO (30 mg/kg) anxiolytic-like effect, as shown by significant increase in time in open arms and ratio of time in open arms and total time in arms as compared to control group, and ZM241385 alone did not show significant changes of all EPM parameters as compared to control group. Caffeine pretreatment: number of open arm entries [F_(4,61)_ = 4.021, *p* < 0.01], time spent in open arms [F_(4,61)_ = 6.074, *p* < 0.0005], ratio time in open arms and total time in arms [F_(4,61)_ = 6.267, *p* < 0.0005]. DPCPX and ZM241385 pretreatment: number of open arm entries [F_(5,76)_ = 2.558, *p* < 0.05], time spent in open arms [F_(5,76)_ = 4.637, *p* = 0.001], ratio time in open arms and total time in arms [F_(5,76)_ = 4.303, *p* < 0.002]. Each bar represents the mean value ± SD. Tukey test: * *p* < 0.05, ** *p* < 0.01, *** *p* < 0.001. Ctrl raw mean values: (**A**) 6.07; (**B**) 55.35 s; (**C**) 0.31; (**D**) 8.44; (**E**) 75.79 s; (**F**) 0.33.

**Figure 4 ijms-21-09281-f004:**
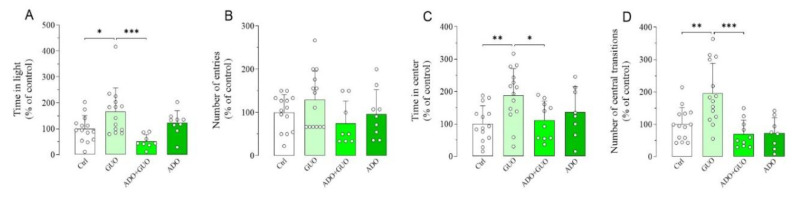
Anxiolytic-like effect of GUO evaluated by light–dark box (LDB) and open field test (OFT) test during a 5 min session. (**A**,**B**) The LDB data analysis revealed that GUO treatment (30 mg/kg) significantly increased the time spent in light box, as compared to control (Ctrl) group, whereas the number of entries although increased did not become significant. This anxiolytic-like effect of GUO treatment was completely blocked by pretreatment with nonselective ARs agonist ADO. ADO treatment alone did not affect behavior as compared to control group. (**C**,**D**) The OFT data analysis revealed that GUO treatment (30 mg/kg) significantly increased both the number of central transitions and the amount of time spent in the center of the arena compared to control group. This anxiolytic-like effect of GUO was completely blocked by pretreatment with ADO, and ADO treatment alone did not affect behavior as compared to control group. LDB: time in light [F_(3,41)_ = 6.146, *p* < 0.002], number of entries [F_(3,41)_ = 1.843]. OFT: time in center [F_(3,42)_ = 4.330, *p* < 0.01], number of central transitions [F_(3,42)_ = 9.930, *p* < 0.0001]. Each bar represents the mean value ± SD. Tukey test: * *p* < 0.05, ** *p* < 0.005, *** *p* < 0.0005. Ctrl raw mean values: (**A)** 26.69 s; (**B**) 2.29; (**C**) 24.29 s; (**D**) 8.86.

**Figure 5 ijms-21-09281-f005:**
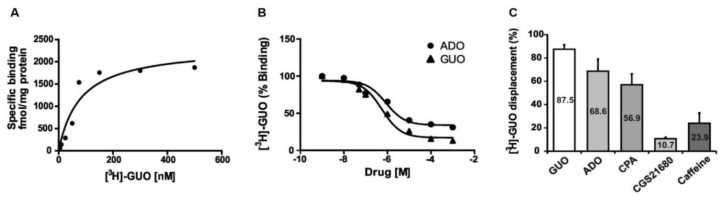
[^3^H]GUO binding to hippocampal membranes. (**A**) The saturation isotherm studies showed that the binding became saturable at [^3^H]GUO concentrations ranging between 100 and 300 nM. The pooled data resolved for the presence of a single high affinity binding site with an apparent K_D_ = 80 ± 34 nM; B_max_= 2339 ± 339 fmol/mg /protein. (**B**) Displacement of [^3^H]GUO binding by GUO and nonselective ARs agonist ADO in rat hippocampal membranes. Competition binding between GUO and ADO showed for ADO almost the same potency order of GUO to displace [^3^H]GUO (pIC_50_ 6.069 ± 0.2074 and pIC_50_ −6.251 ± 0.1649, respectively), although ADO was able to displace only 70% of [^3^H]GUO binding. (**C**) [^3^H]GUO displacement (70 nM) by 500 µM of GUO, ADO, caffeine and selective agonists CPA or CGS21680 in rat hippocampal membranes.ADO was almost as effective as GUO in displacing [^3^H]GUO binding. Selective A_1_R agonist CPA or selective A_2A_R agonist CGS21680 displaced respectively 57% and 11% of [^3^H]GUO binding. Nonselective ARs antagonist caffeine displaced only 24% of [^3^H]GUO binding.

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
