# Peer review of "Guanosine-Mediated Anxiolytic-Like Effect: Interplay with Adenosine A_1_ and A_2A_ Receptors"

_ijms, 2020, doi:10.3390/ijms21239281_

Round 1
Reviewer 1 Report
This paper presents a series of experiments which suggest that guanosine has anxiolytic-like activity in three widely used animal models. Further the study presents evidence that this may occur through a specific guanosine binding site or at least through actions as an allosteric modulator of adenosine sites. The experiments appear to be performed under standard conditions with reasonably clear cut (and statistically significant) results. Data are presented as a series of bar charts as a % of control values which do not allow the reader to examine raw data and therefore assess the results for themselves. This might have been provided as part of the supplementary files. Similarly open field data are not presented (these data are critical for assessing effects of drugs on locomotor activity to rule out sedative effects) which may also have been presented in the supplementary files. In terms of the methodology of the tests employed it is important to report the light intensity at the floor of the apparatus used as excessive light intensity is aversive to rats. Also in the EPM it is usual to report the arm of the apparatus that each animal was facing at the commencement of the behavioural test.
The authors have reported a number of competition experiments with adenosine agonists but only one dose was used in these experiments. It is not clear how these doses were chosen: the basis of previous studies? literature values?
It is curious that caffeine was anxiolytic. In human studies caffeine is often anxiogenic while 'caffeinism' (i.e., excessive consumption of caffeine) is often the cause of anxiety in clinical presentations. To what extent might the results observed here be related to dose of the substance employed? Was a dose response effect explored with caffeine?
Author Response
This paper presents a series of experiments which suggest that guanosine has anxiolytic-like activity in three widely used animal models. Further the study presents evidence that this may occur through a specific guanosine binding site or at least through actions as an allosteric modulator of adenosine sites. The experiments appear to be performed under standard conditions with reasonably clear cut (and statistically significant) results. Data are presented as a series of bar charts as a % of control values which do not allow the reader to examine raw data and therefore assess the results for themselves. This might have been provided as part of the supplementary files.
We thank the Reviewer for these observations. We presented our behavioral data as percentage of the control value (i.e. saline treated animal) for each day experiment. This normalization allowed us to pool in a single graph the results from animals tested in different experiments performed in successive days. Now, we provide, in the figures legends, the raw mean value for the control group in each behavioral test just to give an idea of the real magnitude order of the measurements. In addition, we graph the data as a bars plot showing each data point (animal) for each condition.
Similarly, open field data are not presented (these data are critical for assessing effects of drugs on locomotor activity to rule out sedative effects) which may also have been presented in the supplementary files.
We have now inserted a new supplementary figure (S3A-C) on page 17, showing scatter bar plot of the total distance travelled, the path length in center of the arena and the total mobility time. As already reported in the text, no sedative effects can be detected after acute drug treatment.
In terms of the methodology of the tests employed it is important to report the light intensity at the floor of the apparatus used as excessive light intensity is aversive to rats. Also in the EPM it is usual to report the arm of the apparatus that each animal was facing at the commencement of the behavioural test.
We agree with the Reviewer that the light intensity used for our behavioral experiments should be indicated. Thus, in the new version of the manuscript we reported that, both EPM and OFT, were performed in a mean light intensity (100 lx) illuminated chamber (Page 12, line 330, 337). In addition, we also indicated in the EPM tests the animal faces one of the open arms of the apparatus at the beginning of the test (Page 12, line 329).
The authors have reported a number of competition experiments with adenosine agonists but only one dose was used in these experiments. It is not clear how these doses were chosen: the basis of previous studies? literature values?
Following the reviewer suggestions, we indicated in the new version of the manuscript the criteria of choosing a single dose of adenosine receptors ligands in the competition experiments (Page 13, line 361-364; 367-369). Below more detailed informations.
CPA dose was chosen based on preliminary investigations results. We used first the dose of 0.4 mg/kg, which induced immediate sedative and hypokinetic effects. We also noticed similar sedative effects with the dose of 0.2 mg/kg, accompanied, 45 minute after injection, by the increase in the number of entries in open arms, indicating that this dose of CPA produced per se anxiolytic-like effects. Finally, CPA 0.1 mg/kg, used for the experiments reported in the manuscript, did not produce any anxiolytic-anxiogenic response, in agreement with literature data (Jain, N., Kemp, N., Adeyemo, O. et al. Anxiolytic activity of adenosine receptor activation in mice. Br J Pharmacol., 1995; El Yacoubi, M., Ledent, C., Ménard, J. F. et al. The stimulant effects of caffeine on locomotor behaviour in mice are mediated through its blockade of adenosine A(2A) receptors. Br J Pharmacol., 2000)
CCPA dose (0.1 mg/kg) was chosen based on literature data. According to them, this dose is not anxiolytic (Prediger, R., da Silva, G., Batista, L. et al. Activation of Adenosine A1 receptors reduces anxiety-Like behavior during acute ethanol withdrawal (hangover) in mice. Neuropsychopharmacology, 2006; Prediger, R., Batista, L., Takanashi, R. Adenosine A1 receptors modulate the anxiolytic-like effect of ethanol in the elevated plus-maze in mice. Eur J Pharmacol., 2004), while higher doses (0.25 mg/kg) produce anxolytic-like effects (Prediger, R., da Silva, G., Batista, L. et al. Activation of Adenosine A1 receptors reduces anxiety-Like behavior during acute ethanol withdrawal (hangover) in mice. Neuropsychopharmacology, 2006; Florio, C., Prezioso, A., Papaioannou, A. et al. Adenosine A1 receptors modulate anxiety in CD1 mice. Psychopharmacology, 1998). In agreement, we found that CPA 0.1mg/kg per se does not produce any anxiolytic-anxiogenic response.
The dose of CGS21680 was also chosen according to literature data. There is a large variation in CGS21680 doses. In general, even very low dose (0.05 and 0.1 mg/Kg) of CGS21680 are associated to sedative effects (Mingote, S., Pereira, M., Farrar, A.M., et al. Systemic administration of the adenosine A(2A) agonist CGS 21680 induces sedation at doses that suppress lever pressing and food intake. Pharmacol Biochem Behav. 2008; Karcz-Kubicha, M., Antoniou, K., Terasmaa, A., et al. Involvement of adenosine A1 and A2A receptors in the motor effects of caffeine after its acute and chronic administration. Neuropsychopharmacology, 2003). We opted for CGS21680 0.4 mg/Kg driven by the observation that no significant variations in the EPM parameters were detected for doses ranging between 0.1 and 0.5 mg/Kg, whereas strong evinces of anxiety and hypolocomotion were associated to CPA 2.5 mg/Kg (El Yacoubi, M., Ledent, C., Ménard, J. F. et al. The stimulant effects of caffeine on locomotor behaviour in mice are mediated through its blockade of adenosine A(2A) receptors. Br J Pharmacol., 2000).
It is curious that caffeine was anxiolytic. In human studies caffeine is often anxiogenic while 'caffeinism' (i.e., excessive consumption of caffeine) is often the cause of anxiety in clinical presentations. To what extent might the results observed here be related to dose of the substance employed? Was a dose response effect explored with caffeine?
The Reviewer highlighted and important issue here. Together with a general agreement on caffeine dose-dependent effects, with usually no effects or even anxiogenic effects associated to low dose of caffeine and anxiolytic-like effects related to higher doses, there are some discrepancies in the literature about caffeine modulation of anxiety, even when used at the same dose. For example, Almeida et al. reported that caffeine (10 mg/kg) administration per se did not modulate any anxiety-like behavioral parameters in the EPM task, whereas is able to block GUO-induced anxiolytic like effects (Almeida, R.F., Comasseto, D.D., Ramos, D.B. et al. Guanosine Anxiolytic-Like Effect Involves Adenosinergic and Glutamatergic Neurotransmitter Systems. Mol Neurobiol., 2017). The same dose of caffeine seems to produce a trend of anxiogenic effects, which become significant with caffeine higher doses (25 and 50 mg/kg) (Bhattacharya, S.K., Satyan, K.S., Chakrabarti, A. Anxiogenic action of caffeine: an experimental study in rats. J Psychopharmacol., 1997). Accordingly, caffeine 50 or 100 mg/kg induces anxiogenic effects in both EPM and LDB (El Yacoubi, M., Ledent, C., Ménard, J. F. et al. The stimulant effects of caffeine on locomotor behaviour in mice are mediated through its blockade of adenosine A(2A) receptors. Br J Pharmacol., 2000). Also, caffeine 30mg/kg was anxiolytic in the open field test (Schulz, D. Acute food deprivation separates motor-activating from anxiolytic effects of caffeine in a rat open field test model. Behav Pharmacol. 2018), whereas at a dose of 20 mg/Kg increased the height of buried bedding as well as the duration of burying behavior thus demonstrating a clear anxiogenic action (Vitale, G., Filaferro, M., Ruggieri, V., et al. Anxiolytic-like effect of neuropeptide S in the rat defensive burying. Peptides, 2008). Given literature heterogeneity, we tried in preliminary investigations two different doses of caffeine (10 and 30 mg/Kg). In our experimental model, both doses were anxiolytic and produced per se an increase in the time spent in open arms in the EPM test. Given the total overlapping of results between the two doses, we showed in the manuscript the effect of the higher dose (30mg/Kg) only. We can’t exclude here that caffeine pretreatment, in combination with GUO, might interfere with GUO effects, since no additive anxiolytic effects between the two molecules were observed. However, we are enough confident in excluding antagonistic effects of caffeine towards GUO, since caffeine did not efficiently antagonize GUO binding and pre-treatment with specific ADO receptor antagonists, which per se do not show anxiolytic nor anxiogenic properties, did not interfere with GUO effects.
Reviewer 2 Report
The manuscript entitled “Guanosine-mediated anxiolytic-like effect: interplay with adenosine receptors” focuses on the validation of A1 and A2A receptors in the anxiolytic-like effects of guanosine (GUO) using a model of anxiolytic-like effect induced by acute guanosine treatment in naïve rats.
The manuscript is well written, it has a complete experiment design, and the discussion is detailed and correct. The results obtained are clear. However, I have some comments:
-The title of the manuscript “Guanosine-mediated anxiolytic-like effect: interplay with adenosine receptors” may be changed in “Guanosine-mediated anxiolytic-like effect: interplay with A1 and A2A adenosine receptors”
-GUO may bind to A1 and A2A receptors in competitive manner with adenosine. The authors found that the anxiolytic-like effect of GUO can be blocked by A1R and A2AR agonists but not by A1R and A2AR antagonists. Have you any data about adenosine levels in relation to GUO effect?
-Please revise Figure 4 C since the title of y axis is not clear to the reader.
Author Response
The manuscript entitled “Guanosine-mediated anxiolytic-like effect: interplay with adenosine receptors” focuses on the validation of A1 and A2A receptors in the anxiolytic-like effects of guanosine (GUO) using a model of anxiolytic-like effect induced by acute guanosine treatment in naïve rats.
The manuscript is well written, it has a complete experiment design, and the discussion is detailed and correct. The results obtained are clear. However, I have some comments:
-The title of the manuscript “Guanosine-mediated anxiolytic-like effect: interplay with adenosine receptors” may be changed in “Guanosine-mediated anxiolytic-like effect: interplay with A1 and A2A adenosine receptors”
We thank the referee for the appreciation. We follow the reviewer’s recommendation and we modify the title accordingly.
-GUO may bind to A1 and A2A receptors in competitive manner with adenosine. The authors found that the anxiolytic-like effect of GUO can be blocked by A1R and A2AR agonists but not by A1R and A2AR antagonists. Have you any data about adenosine levels in relation to GUO effect?
The reviewer raised an important question here. Indeed, it can’t be ruled out that GUO-mediated effects might be related to alterations in adenosine levels, however, we do not have data regarding this contention. Thus, future experiments will be designed to answer this important question. However, it is known that GUO treatment is able to produce a release of ADO in cultured astrocytes (Di Iorio, P., Kleywegt, S., Ciccarelli, R. et al. Mechanisms of apoptosis induced by purine nucleosides in astrocytes. Glia, 1999) and to increase extracellular adenosine by modifying its disposition in vascular and endothelial cell cultures (Jackson, E.K., Cheng, D., Jackson, T.C., et al. Extracellular guanosine regulates extracellular adenosine levels. Am J Physiol Cell Physiol., 2013; Jackson, E. K., Gillespie, D.G. Regulation of Cell Proliferation by the Guanosine-Adenosine Mechanism: Role of Adenosine Receptors. Physiol reports, 2003). However, the observation that pre-treatment with adenosine antagonists does not inhibit GUO-induced anxiolytic-like effects seems to exclude, at least in this specific case, the involvement of an adenosine increase so great to interfere with GUO effects, while supporting a primary role of GUO. If anything, ADO and related agonists are able to displace GUO binding, while preserving their ability to bind their own receptors and by this way to determine their anxiogenic effects.
-Please revise Figure 4 C since the title of y axis is not clear to the reader.
We thank reviewer for this observation, and we apologize for the inconvenience. Thus, we amended accordingly the figure. Now the y axis is titled “number on central transitions (% of control).